# Modelling Neuromuscular Diseases in the Age of Precision Medicine

**DOI:** 10.3390/jpm10040178

**Published:** 2020-10-17

**Authors:** Alfina A. Speciale, Ruth Ellerington, Thomas Goedert, Carlo Rinaldi

**Affiliations:** Department of Paediatrics, University of Oxford, Oxford OX1 3QX, UK; ambra.speciale@paediatrics.ox.ac.uk (A.A.S.); ruth.ellerington@paediatrics.ox.ac.uk (R.E.); tg5g18@soton.ac.uk (T.G.)

**Keywords:** neuromuscular diseases, translational research, disease models, precision medicine

## Abstract

Advances in knowledge resulting from the sequencing of the human genome, coupled with technological developments and a deeper understanding of disease mechanisms of pathogenesis are paving the way for a growing role of precision medicine in the treatment of a number of human conditions. The goal of precision medicine is to identify and deliver effective therapeutic approaches based on patients’ genetic, environmental, and lifestyle factors. With the exception of cancer, neurological diseases provide the most promising opportunity to achieve treatment personalisation, mainly because of accelerated progress in gene discovery, deep clinical phenotyping, and biomarker availability. Developing reproducible, predictable and reliable disease models will be key to the rapid delivery of the anticipated benefits of precision medicine. Here we summarize the current state of the art of preclinical models for neuromuscular diseases, with particular focus on their use and limitations to predict safety and efficacy treatment outcomes in clinical trials.

## 1. Introduction

Neuromuscular diseases are a broad and heterogeneous group of conditions characterized by an impairment in one or more components of the motor unit, defined as the motor neuron and the muscle fibres it innervates. Whilst most are individually rare, collectively neuromuscular diseases are significantly prevalent, with a cumulative prevalence of approximately 100–200 cases per 100,000 individuals worldwide [1], accounting for a substantial proportion of population-wide health care costs [2]. Very few treatments currently exist to treat these diseases. Nevertheless, as research progressively disentangles their pathogenic mechanisms, many opportunities are finally starting to land in the clinic.

Precision medicine refers to a treatment approach wherein the most appropriate treatment for an individual is chosen based on their specific disease manifestation, alongside their genetic/epigenetic information and other features such as their microbiome, age, nutrition, and lifestyle. The clinical and genetic heterogeneity of neuromuscular diseases make them ideal candidates for personalized therapeutic approaches, with many individuals suffering from rare or ultrarare diseases that cannot be treated by conventional blanket approach treatment. One example is Duchenne muscular dystrophy (DMD), the most prevalent childhood-onset muscular dystrophy, where progressive muscle degeneration and weakness is caused by mutations in the *DMD* gene, leading to loss of dystrophin protein production [3]. The vast majority of DMD patients carry an exon deletion (~65%) or a duplication (~10%) of one or multiple exons and these mutations tend to manifest in regions of vulnerability between exons 2 and 20 and exons 45 and 55 [4,5,6]. In addition, small mutations (insertions, deletions, nonsense mutations and splice site mutations) account for the remaining ~25% mutations and occur throughout the length of the gene [4]. Excision of specific exons, or exon skipping, by use of antisense oligonucleotides (AON) to allow restoration of the disrupted reading frame and therefore production of a shortened but functional dystrophin protein, has surfaced as a promising therapy for DMD [7]. Therefore, diagnosis by genetic sequencing has become a crucial tool in determining eligibility for these treatments, as multiple AON products need to address the large series of mutations carried by DMD subjects.

While presenting new challenges for researchers, precision medicine is rapidly taking the lead in the pursuit of radically transforming health care. Choosing the appropriate disease model that recapitulates the complexity and heterogeneity of patients is therefore paramount to understand disease mechanisms and increase the chances of success of translating a treatment opportunity into a safe and effective marketed drug.

In this review, we aim to discuss the currently available tools used to model neuromuscular diseases and to evaluate their utility and applicability to personalized medical research and therapeutic development (Table 1).

## 2. Cellular Models

### 2.1. Myoblasts

Primary myoblasts (activated satellite cells) obtained from human subjects or animal models typically go through multiple rounds of cell division until reaching confluence in growth media, followed by iterations of cellular fusions to form multinuclear myotubes and eventually terminal differentiation [8]. Due to several inherent traits of human-derived muscle cells, including the slower growth rate as well as the flattened morphology, primary human myotubes typically exhibit poorer contractile activity than their mouse counterparts in response to electric stimulation [9]. Obtaining a substantial number of satellite cells from skeletal muscle biopsies of patients is markedly limited by the restricted proliferative capability of activated satellite cells in culture. In order to overcome this limitation, myogenic conversion of non-muscle primary cells, such as primary human and murine fibroblasts from skin, has been widely employed, mainly using transduction of *MyoD* gene (myogenic differentiation), a master regulator of skeletal muscle differentiation [10]. In order to increase proliferative capacity, transduction with both telomerase-expressing and cyclin-dependent kinase 4-expressing vectors has been used to produce immortalized human muscle stem-cell lines from patients with different muscle diseases such as DMD, limb-girdle muscular dystrophy type 2B, facioscapulohumeral muscular dystrophy, oculopharyngeal muscular dystrophy and congenital muscular dystrophy [11]. These immortalized cultures have been extensively used both to study disease mechanism and to test treatment strategies.

### 2.2. Induced Pluripotent Stem Cells (iPSCs)

The development of induced pluripotent stem cell (iPSC) technology has brought a great paradigm shift in the field of precision medicine [12] and now they have a prominent role as a tool for disease modelling and drug screening. Moreover, they are highly expandable, are free from the ethical issues linked to the use of embryonic stem cells (ESCs), and their source of cells easily accessible.

Two major strategies have been recently developed to differentiate PSCs into satellite-like cells. The first involved overexpressing PAX7, the master transcription factor for satellite cells, in an inducible fashion [13]. After being generated from human embryonic stem cells and iPSCs, these cells showed capability for in vitro expansion and differentiation, as well as engraftment and myofibre formation in immunodeficient mice [13,14]. The second strategy involved the use of a small molecule, and consists of glycogen synthase kinase 3 beta (GSK3beta) inhibition, in order to activate the Wnt pathway, as well as treatment with fibroblast growth factor 2 (FGF2) in a minimal medium [15,16,17,18,19,20]. Alternative protocols have used bone morphogenic protein 4 (BMP4) inhibition to promote differentiation into the myogenic lineage [21,22,23], or Notch signalling inhibitor DAPT [24]. Purified by fluorescence-activated cell sorting (FACS) [15,19,24], partially purified, or unpurified [16,17,20,21,23], cell mixtures are then plated.

By generating an in vitro DMD model from patient-derived iPS cells, Shoji et al. noted excess Ca^2+^ influx in DMD myocytes when compared to control myocytes in response to stimulation via electricity. This was alleviated by restoring dystrophin expression via exon skipping, therefore establishing a model that recapitulates early DMD pathogenesis and is appropriate for assessing the efficacy of exon-skipping drugs by phenotypic assay [25]. IPSC models of several other neuromuscular diseases are currently available, including Miyoshi myopathy, a muscle disease caused by the mutation in dysferlin [26], Pompe disease, a paediatric disease caused by lysosomal glycogen accumulation in skeletal muscle that leads to muscle weakness [27], and myotonic dystrophy type 1, a multisystem disorder that affects skeletal and smooth muscle caused by a CTG trinucleotide repeat expansion in the non-coding region of the *DMPK* gene [28]. Overall, the introduction of iPSC technology has allowed scientists to model diseases directly from patients’ cells, this being a cornerstone for personalized medicine. However, if they are planned to be used for personalized cell therapy, several issues remain to be addressed, including alterations in the differentiation efficiency, line-to-line variability, and risk of tumorigenicity.

### 2.3. Urine-Derived Stem Cells

In addition to representing an ideal source of cells for generating iPSCs, with a reprogramming efficiency approximately 100-fold higher than that of fibroblasts [29], urine stem cells (USCs) can also be induced into myogenic lineage by direct MyoD1 reprogramming [30]. Muscle differentiation can be further enhanced by adding 3-deazaneplanocin A hydrochloride [31]. These cells carry pluripotency markers such as CD29, CD105, CD166, CD90, and CD13 [32], and are able to self-renew and differentiate into the mesodermal, endodermal and ectodermal lineage [33]. Direct reprogramming of these cells, which can be easily isolated by centrifugation method and standard cell culture, has been recently shown to efficiently and reproducibly establish human myogenic cells from patients with DMD and limb-girdle muscular dystrophy (LGMD) type 2 [30]. Upon further molecular characterisation, this cost-effective and efficient in vitro model system shows great potential for more efficient drug development and targeted therapies development for neuromuscular diseases.

### 2.4. Skeletal Muscle Organoids

As the use of human iPSCs for tissue engineering and disease modelling expands, iPSC-derived organoids are rapidly becoming a powerful tool for modelling human organogenesis, homeostasis, injury repair and disease aetiology [34]. These miniature 3D tissues are generated using a combination of signposted differentiation, morphogenetic processes, and the embryonic organogenesis mimicking intrinsically driven self-assembly of cells, resulting in architecture and function remarkably similar to their in vivo counterparts. By using natural or synthetic scaffolds to create the artificial tissue [35], these models account for the cell–cell and cell–extracellular matrix interactions as well as the mechanical and/or chemical cues [36,37]. The development of physiologically relevant 3D in vitro models holds great promise to provide more economic, scalable and reproducible means of testing drugs and therapies for successful clinical translation. Few studies have reported methods to engineer human skeletal muscle tissue [38,39,40,41,42,43]. Induced myogenic progenitor cells derived from multiple human iPSC lines have been shown to form functional skeletal muscle tissues and are able to survive, progressively vascularize, and maintain functionality when implanted into the hindlimb muscle or dorsal window chamber in immunocompromised mice [44]. Isogenic human iPSC-derived 3D artificial muscles from patients affected by DMD, limb-girdle type 2D, and lamin A/C (LMNA)-related muscular dystrophies have been recently generated, recapitulating several pathogenic hallmarks in these diseases and also showing potential for muscle engraftment [45]. These studies have indicated that generation of fully functional artificial muscles require the contribution from other cellular lineages, for example vascular cells and motor neurons [45,46,47,48,49]. The major challenges the field is currently facing are mainly related to improving organoids’ scalability as well as their complexity and maturity. Recent success in growing brain organoids using multiwell spinning bioreactors represents a significant step towards high-throughput drug screening via large-scale organoid generation [50]. These models resemble more closely foetal than adult tissue, therefore optimisation of protocols is essential before being able to advance these tissues into replacement therapy. Bearing in mind the speed at which the field has advanced over the past few years, the range of possible future applications of this platform in the study of human diseases and in regenerative medicine is expected to rapidly expand.

### 2.5. Muscle on Chip

Advancement in culturing models with mixed culture capabilities, together with the latest developments in 3D printing, microfluidics and microfabrication engineering, has led to the rapid expansion of organ-on-chip technologies. These platforms have recently attracted substantial interest due to their potential to be informative at multiple stages of the drug discovery process, while offering new ways to model disease states and perform mechanistic investigations in vitro. The critical and defining features of these platforms are the 3D structure, the possibility of integration of multiple cell types to reflect tissue physiology, and the presence of relevant biomechanical forces [51]. Organ on chips have been adapted for the human gut [52], heart [53], blood–brain barrier [54], and kidney [55]. Human primary myogenic cells have been engineered to form 3D myobundles, which respond to electrical stimuli and undergo dose-dependent hypertrophy or myopathy in response to pharmacological stimulation [40]. The decreased muscle regeneration capacity and weakness observed in DMD patients have been recapitulated in a human dystrophic skeletal muscle on a chip [56]. Using a 3D photo-patterning approach, other researchers have developed a skeletal muscle platform by confining a cell-laden gelatin network around two hydrogel pillars, which serve as anchoring sites for the cells, as the muscle tissues form and mature [57]. In other instances, neurons and rhabdomyocytes, both originating from mouse embryonic cells, have been differentiated in a 3D hydrogel culture, to effectively constitute a neuromuscular unit on a chip [58].

Tissue engineering requires a deep understanding of the functional interplay of cell types and the effect of the scaffold on cellular architecture, as well as careful characterising and validation of the model for the purpose of study. Additionally, due to safety concerns around the potential for unexpected toxic side effects, the biocompatibility of the materials to be used must be well profiled [51].

As iPSCs or adult stem cells taken from mass production of tissue organoids are increasingly employed as a source of cells for these platforms, organ on a chip represents an ideal tool for precision medicine.

### 2.6. Other

Sources in addition to the muscle-derived cells or reprogrammed cells can be employed to model muscle diseases. For example, melanocytes from DMD patients show the same morphological alterations as DMD muscle-derived cells [59]. Cultured melanocytes from skin biopsies have been shown to be a useful alternative to muscle biopsies for the mRNA-based molecular diagnosis of DMD [60]. Additionally, in the case of Ullrich congenital muscular dystrophy (UCMD) and Bethlem myopathy (BM), diseases caused by mutations in collagen VI genes [61], patients’ derived melanocytes recapitulated the mitochondrial dysfunction and ultrastructural alterations that are found in patient myoblasts [62].

## 3. Animal Models

### 3.1. Mouse Models

A large fraction of currently available therapies have been developed with the help of animal models, especially mice, mainly due to the high similarity in sequence homology and organ physiology to humans, as well as cost-effective husbandry. Additionally, the external environment in mice studies can be well controlled and monitored and studies using inbred mice allow resampling isogenic individuals, therefore minimising variability.

Nevertheless, many differences remain: mice are smaller in size, have a markedly reduced lifespan and an increased heart rate, just to name a few. Approximately 1% of human genes are not present in the mouse genome [63], while the differences in the promoter regions, non-coding sequences, and RNA splicing might be even more marked, accounting for species-specific disparities in gene expression that in some cases can affect disease phenotype [64,65]. Overall these considerations, together with the realisation that treatments in mice have frequently resulted in disappointing outcomes in clinical trials, have recently called into question the translational potential of findings in mouse models [66].

One way of making mouse models for studying human diseases more suitable is to follow approaches pioneered over 30 years ago, which comprise incorporating human DNA into the mouse genome (genetic humanisation) and/or engrafting human cells and tissue into mouse tissues (cellular humanisation) [67,68,69,70]. Genetic humanisation can be achieved through a variety of methods, most commonly by injection of plasmids or artificial chromosome vectors into the mouse zygotes. Transgenic models have substantially contributed to advancing the understanding of human disease and have helped develop treatment strategies. One notorious major breakthrough in biomedical research using transgenic mice carrying the human *SMN2* gene led to the recent clinical approval of an AON, able to block an intronic splicing silencer in human *SMN2* [71], increasing full-length *SMN2* isoform expression, which compensates for the loss of *SMN1* that causes spinal muscular atrophy [72,73,74,75].

However, some key features must be considered: the cDNA or genomic DNA used to generate the transgenic mice tend to integrate randomly in multiple copies and thus overexpress the protein of interest. Overexpression of wild-type proteins may give a dose-dependent phenotype not related to the disease mutation, like in the case of the androgen receptor [76], and RNA binding proteins, such as TAR DNA-binding protein 43 [77]. The rise of genome engineering technology has revolutionized the field of molecular biology by allowing the generation of physiological, humanized knock-in mice models by precise editing [78,79]. Most DMD preclinical studies have been carried out in the mdx mouse that carries a nonsense point mutation in DMD exon 23 [80], which is only one out of the thousands of possible variations in this gene present in DMD patients. Despite a lack of dystrophin expression, these mice do not exhibit dilated cardiomyopathy or a shortened lifespan. To improve upon this model, a number of double knock-out mouse models have been created, such as mice deficient in both dystrophin and its homolog utrophin, which show decreased cardiac function and survival [81]. In recent years by using clustered regularly interspaced short palindromic repeat (CRISPR)-based editing, many new DMD mouse models carrying deletions, frameshifting mutations, a point mutation, and a mutant version of the human *DMD* gene have been generated [82,83,84,85,86,87,88], making testing of exon skipping strategies targeting different parts of the DMD transcript possible. It is worth considering that recent studies to assess the effects of disease-causing mutations or environmental stimuli in different mouse strains found a strong influence of the genetic background on phenotypic responses [89], highlighting the importance of genetic diversity of animal models in biomedical research.

It is becoming more and more evident that choosing the right model is critical. Depending on the specific research question, often combining different strains is the most appropriate way to minimize the risks of a lack of reproducibility of translational research. Despite the obvious differences between mice and humans, genetic mouse models have allowed us to look at the effects of a mutation at a system level. Combining genetic engineering, which has made genetic modifications of endogenous targets possible, with the use of genetic with cellular humanisation, we now have powerful tools to study human pathophysiology in vivo, in cell-autonomous and non-cell-autonomous contexts [90], as well as excellent preclinical models to identify and test the pharmacodynamic and pharmacokinetic properties of a treatment strategy, from gene therapy to small-molecule and cell replacement [91]. Overall, these considerations further support the use of ‘mouse precision medicine’ as a better prototype for future mouse studies.

### 3.2. Drosophila Melanogaster

Drosophila melanogaster can serve as a useful model of human neuromuscular disease, since flies have a neural circuitry, albeit much simpler than in humans, as well as multinucleated muscle cells and neuromuscular junctions (NMJ). The mechanisms of synaptic transmission seen at the NMJ in humans are conserved in Drosophila, with a key difference being that Drosophila uses glutamate, not acetylcholine, as the neurotransmitter. The ability to genetically manipulate Drosophila is useful when trying to better understand how certain myopathies occur. Moreover, their short life span and large progeny make flies a good system for carrying out large-scale genetic screens. Drosophila has helped us understand more about the NMJ, and in particular, the role that the dystrophin–glycoprotein complex plays (DGC). Like in mammals, the Drosophila gene of dystrophin also encodes multiple isoforms, which contain highly conserved domains and are mainly expressed in the muscle and the nervous system [92,93,94]. Studies into DGC function at the NMJ of Drosophila have shown that it plays an important role in the retrograde control of neurotransmitter release, neuronal migration and muscle stability and thus may help explain how neuromuscular pathology can occur. Removal of a dystrophin isoform (DLP2) in Drosophila, which is normally located at the post-synapse, has been shown to lead to an increase in presynaptic neurotransmitter release, causing increased muscle depolarisation, thus indicating a role of dystrophin in regulating presynaptic neurotransmitter release [95]. Previous work has shown that by studying sensory neurons (photoreceptor cells) in Drosophila [96], a lot can be learnt about axon guidance and target recognition. Perturbation of dystrophin and dystroglycan in photoreceptor cells led to disrupted axon guidance, similar to neuronal defects seen in human muscular dystrophy patients. Drosophila not only aids us in understanding the role that certain proteins play at the synapse of the NMJ, but also serves as a good model for studying age-dependent progression of muscular dystrophy. The reduction in levels of expression of dystrophin isoforms in Drosophila using RNAi led to muscle degeneration in larval and adult flies [95], thus potentially providing a useful model to help us understand Duchenne muscular dystrophy pathogenesis in humans.

### 3.3. Zebrafish

The zebrafish (Danio rerio) has become a useful organism for studying neuromuscular genetic disorders [97]. Comparison to the human reference genome has shown that approximately 70% of human genes have at least one zebrafish orthologue [98], and dozens of mutant zebrafish lines have already been generated to model the most common human myopathies [99,100,101]. As vertebrates, they possess desirable attributes, including small size, rapid development, and genetic tractability [97]. Zebrafish embryos are transparent, develop externally and can be easily genetically manipulated [102], making this model ideal for phenotypic high-throughput screening platform to investigate drug efficacy in a whole-organism context. The most commonly adopted screening criteria for assessing neuromuscular phenotype are spontaneous coiling, ability to hatch on time, swimming behaviour, and birefringence assay [103]. Compared to target-based drug discovery, a phenotype-driven approach offers several key advantages [104], such as rapid identification of compounds that have poor bioavailability, exhibit toxicity or off-target effects. By screening small-molecule libraries in the dystrophin-null zebrafish (sapje model), aminophylline, a non-selective phosphodiesterase inhibitor, was found to improve survival rate in animals, restore normal muscle structure and up-regulate the cAMP-dependent PKA pathway without affecting dystrophin expression [105]. In the sapje model, the mitochondrial defects present in DMD patients were recapitulated, making it an optimal model for the disease, and it was used to assess the effect of the cyclophilin inhibitor alisporivir treatment in vivo, resulting in an improvement in the morphology of mitochondria and myofibrils, and in mitochondrial respiration [106]. A zebrafish model showing severe myopathy has also been generated for UCMD via a deletion in the *col6a1* gene through the injection of an antisense morpholino [107]. Here, defects in the mitochondria permeability transition pore (mPTP) were corrected with the cyclophilin inhibitor NIM811 treatment [108]. In another study, the zebrafish model was used to test mitochondrial respiratory capacity after treatment with stable analogues of mPTP inhibitors [109]. Additionally, the zebrafish model has also provided insight into functional aspects of disease pathogenesis for several muscle conditions: for example, studies in zebrafish relatively relaxed (ryr) mutant, a model of RYR1-related myopathies [110], have contributed to identifying oxidative stress as an important disease mechanism in RYR1-related myopathies [111].

### 3.4. Caenorhabditis Elegans

With 40% of human disease genes having a nematode ortholog [112], and a fully sequenced genome [113], C. elegans is a valuable model to investigate several human physiological and pathological mechanisms. Studies of sarcomere maintenance and function in striated muscle led to the first identification of many conserved proteins, including twitchin, unc-89 (obscurin), unc-112 (kindlin), unc-45 (myosin chaperone) and unc-78 (AIP1) [114]. Using a large-scale screens in a C. elegans model of muscular dystrophy, carrying mutations in the dys-1 and the hlh-1 genes, which are respectively the homolog for the mammalian dystrophin and *MyoD* gene [115], compounds such as prednisone and serotonin have been shown to be effective in reducing muscle degeneration [116,117]. The obvious advantages of using this scalable and high-throughput model are counterbalanced by the limited phenotypic analyses, such as counting the number of times a worm bends in a C-shaped fashion in liquid in one minute, although new automated methods of quantifying muscle contraction and relaxation kinetics are emerging [118].

## 4. Computational Models

In silico models are becoming an increasingly useful tool for investigating muscle function and in helping us to understand which key players cause muscle pathology. These models integrate published experimental data, thus allowing us to encompass the many variables linked to pathology in a single model, enabling the study of multifaceted diseases. In doing these studies, one may understand better the underlying interactions between different disease mechanisms that lead to pathology, which may prove harder to do in live experiments. Over the last twenty years, big steps have been made in the computational modelling of muscle. A recent development has been the creation of agent-based models (ABMs), which allow us to assess what roles different biological agents play in muscle pathology, both at cellular and systems levels. For example, the use of ABMs for DMD has indicated a link between low satellite stem cell counts and impaired muscle regeneration symptom [119]. ABMs can also be used to predict the outcomes of given scenarios based on the rules derived from the literature, as well as having certain parameters that cannot be measured experimentally. This system can even add software agents that mimic certain biological cells into the simulation, with the aim of helping us to better understand their cellular interactions. This has been carried out in studies showing that fibroblasts can affect a muscle’s susceptibility to disuse-induced atrophy [120].

However, these models do have their limitations: the simulated model is not a full replicate of the muscle cell and its microenvironment, as it only accounts for the contribution of known variables, which renders this model system not fully translatable to the in vivo situation.

## 5. Conclusions

The increasing availability of genetic and phenotypic information on patients with neuromuscular diseases, coupled with the unprecedented opportunity to manipulate eukaryotic genomes to generate disease models to study these diseases, has the potential to accelerate the translation of new therapeutic opportunities from preclinical settings into medical practice. Among the models available to researchers, 3D cultures and muscle on chips are best suited for precision medicine applications, due to their structural complexity and opportunity for genetic and environmental manipulation. However, as it becomes increasingly evident that we need to abandon the concept of ‘one drug fits all’, modelling every disease-associated variant for preclinical applications is likely to be unattainable and in many cases unnecessary. Achieving model precision is critical in translational research as long as it provides predictive validity, which is the ultimate goal of preclinical work, and may further be enhanced by using multiple models to capture the spectrum of mechanisms and testing therapies in diverse genetic backgrounds that more closely reflect the human population as a whole. This may be particularly true in complex diseases, where multiple risk loci concur to the development of a specific condition or to the treatment response.

## Figures and Tables

**Table 1 jpm-10-00178-t001:** Key features of the various models used for neuromuscular diseases.

Models.	Myoblasts	Stem Cells Derived Cultures	Organoids	Muscle-on-a-Chip	Mouse Models	Other Animal Models	Computational Models
**Parameter**	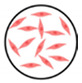	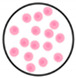	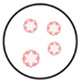	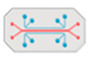	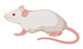	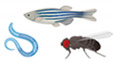	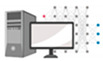
**Production complexity**	Low	Medium, depends on protocol	Medium/High	High, requires engineered chambers	High	High/Medium	Medium/High
**On-platform assay**	Easy access to readouts, individual cell analyses possible	Medium difficulty, individual cell analyses possible	Medium/High difficulty, analyses possible at the tissue level	Medium/High, organ function analyses possible	Low difficulty, analyses possible both at cellular and tissue levels	Low difficulty, analyses possible both at cellular and tissue levels	Can account for contributions only of known variables
**Duration of experiments**	Minutes to days	Days to weeks	Days to weeks	Days to weeks, depends on platform design	Weeks to months	Days to weeks	Days
**Variability and clinical relevance**	Low variability and relevance	High variability and relevance	High variability and relevance	Low variability, high relevance	Low variability, high relevance	Low variability, moderate relevance	Low variability but requires in vivo confirmation
**Level of control over variables**	High	Medium/Low	Low	High	High/Medium	High/Medium	High
**Biodistribution and toxicology studies**	Useful for initial toxicology	Useful for initial toxicology	Useful for initial toxicology and limited biodistribution studies	Useful for initial toxicology and limited biodistribution studies	Useful for toxicology, biodistribution and life span assays	Useful for toxicology and life span assays	Useful to predict outcomes
**High throughput feasibility**	High	Medium/High	Medium/High, depends on tissues	Medium, depends on platform design	Low	High	High
**Precision medicine potential**	Low, easy to manipulate but homogenous cultures	High, takes into account individual variability	High, depends on tissues and is subject to model validation	High, allows high level of control and personalization	High, depends on model availability	Medium/High, easy to manipulate but low translational relevance	High when used in combination with in vitro/vivo methods

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
