# Peer review of "Modelling Neuromuscular Diseases in the Age of Precision Medicine"

_jpm, 2020, doi:10.3390/jpm10040178_

Round 1

Reviewer 1 Report

With the current review article (Manuscript ID: jpm-959607), Alfina, Ruth, and colleagues summarise preclinical models for neuromuscular diseases and highlight use and limitations to predict safety and efficacy treatment outcomes in clinical trials. This is an interesting review, scientifically sound, and fits best for the publication by the Journal of Personalized Medicine. The information included in the manuscript is relevant and systematic. I have only some minor comments.

  1. The paper format does not comply with the Journals’ guidelines.
  2. Place table caption above the Table.
  3. Line 82: write the full name of GSK3beta.
  4. Line 89: myocytes
  5. Italicize Caenorhabditis elegans as Caenorhabditis elegans / elegans throughout the paper.

Author Response

Please find our responses in bold. 

  • The paper format does not comply with the Journals’ guidelines. The paper has now been formatted following the Journal's guidelines.
  • Place table caption above the Table. Caption placed above the table 
  • Line 82: write the full name of GSK3beta. Changed accordingly
  • Line 89: myocytes Changed accordingly
  • Italicize Caenorhabditis elegans as Caenorhabditis eleganselegans throughout the paper. Caenorhabditis elegans, Drosophila and Danio rerio have been italicized throughout the paper

Reviewer 2 Report

Speciale, et al., wrote a potential masterpiece review clarifying almost all the available model to study neuromuscular disease spanning from cellular models to animal models.

Strenght:

1- manuscript is written in a really professional english language and is really understandable;

2- All models necessary to go towards precision medicine are clearly described as well as the possible alternatives to animal models which use actually remains necessary to explain moe complex mechanisms and to asses drug toxicology.

Weakness:

1- I suggest the authors to stress the importance to go towards precision medicine and scientific impact of using all described models to reach the objective.

2- What are the best models according authors? You have to personalize the review conclusions a bit more.

3- I suggest to amplify Cellular models section by inserting these papers about the use of melanocytes isolated from patients with Ullrich congenital muscular dystrophy:

Zulian, et al., 2014, Frontiers Aging Neuroscience - Melanocytes from Patients Affected by Ullrich Congenital Muscular Dystrophy and Bethlem Myopathy have Dysfunctional Mitochondria That Can be Rescued with Cyclophilin Inhibitors.

Pellegrini, et al., 2013, Journal Cell Physiology - Melanocytes--a novel tool to study mitochondrial dysfunction in Duchenne muscular dystrophy.

4 - I suggest to amplify zebrafish section by inserting these papers about the use of zebrafish to study mitochondrial dysfunction at the basis of muscle fiber death in both Duchenne and Ullrich dystrophies, and the use of drugs recovering this defect:

Schiavone, et al., 2017, Pharmacological Research - Alisporivir rescues defective mitochondrial respiration in Duchenne muscular dystrophy.

Zulian, et al., 2014, HMG - NIM811, a cyclophilin inhibitor without immunosuppressive activity, is beneficial in collagen VI congenital muscular dystrophy models.

Sileikyte, et al. 2019, ChemMedChem - Second-Generation Inhibitors of the Mitochondrial Permeability Transition Pore with Improved Plasma Stability.

Author Response

Please find our responses in bold:

  1. I suggest the authors to stress the importance to go towards precision medicine and scientific impact of using all described models to reach the objective. 

    In lines 49-53, we stated that precision medicine will transform healthcare, therefore it is important to choose appropriate disease models to develop effective treatments.  

    2- What are the best models according authors? You have to personalize the review conclusions a bit more. 

    We emphasized that muscle-on-chip and 3D cultures are more suitable for personalized medicine applications. Changes are in lines 322-325 

    3- I suggest to amplify Cellular models section by inserting these papers about the use of melanocytes isolated from patients with Ullrich congenital muscular dystrophy: 

    Zulian, et al., 2014, Frontiers Aging Neuroscience - Melanocytes from Patients Affected by Ullrich Congenital Muscular Dystrophy and Bethlem Myopathy have Dysfunctional Mitochondria That Can be Rescued with Cyclophilin Inhibitors. 

    Pellegrini, et al., 2013, Journal Cell Physiology - Melanocytes--a novel tool to study mitochondrial dysfunction in Duchenne muscular dystrophy. 

    We have introduced a new paragraph under cellular models entitled ‘Other’ (lines 171-178). The references have been added explaining the existence of other cell sources to model muscle diseases such as Duchenne muscular dystrophy, Ullrich Congenital Muscular Dystrophy and Bethlem Myopathy.   

    4 - I suggest to amplify zebrafish section by inserting these papers about the use of zebrafish to study mitochondrial dysfunction at the basis of muscle fiber death in both Duchenne and Ullrich dystrophies, and the use of drugs recovering this defect: 

    Schiavone, et al., 2017, Pharmacological Research - Alisporivir rescues defective mitochondrial respiration in Duchenne muscular dystrophy. 

    Zulian, et al., 2014, HMG - NIM811, a cyclophilin inhibitor without immunosuppressive activity, is beneficial in collagen VI congenital muscular dystrophy models. 

    Sileikyte, et al. 2019, ChemMedChem - Second-Generation Inhibitors of the Mitochondrial Permeability Transition Pore with Improved Plasma Stability. 

    In lines 273-282, we have inserted the references explaining the applicability of the zebrafish model of Duchenne muscular dystrophy and Ullrich Congenital Muscular Dystrophy to study mitochondrial dysfunction and the treatments with mPTP inhibitors.